# Integrating AAA Care Coordinators in Primary Care for Dementia Support: Implementation Challenges and Lessons Learned

**DOI:** 10.3390/ijerph22040506

**Published:** 2025-03-26

**Authors:** Mary C. Ehlman, Suzanne Leahy, Reagan Lawrence, Della Evans

**Affiliations:** 1Bronstein Center for Healthy Aging and Wellness, Kinney College of Nursing and Health Professions, University of Southern Indiana, 8600 University Boulevard, Evansville, IN 47712, USA; sleahy@usi.edu (S.L.); rwlawrence_se@usi.edu (R.L.); 2Deaconess Geriatric Fellowship Program, Deaconess Hospital, Inc., 600 Mary St, Evansville, IN 47747, USA; della.dillard@deaconess.com

**Keywords:** dementia, early detection and diagnosis, comprehensive assessment, community-based support and services, caregiver support, provider education

## Abstract

Patient, provider, and community barriers challenge dementia diagnosis and management in primary care. Interventions emphasizing EHR-based workflows with minimal provider training are insufficient to address these challenges. To improve early detection, dementia care, and the global health of caregivers and patients living with dementia, interventions must take a more comprehensive approach, addressing provider education and helping families be aware of the community supports available. Methods: Through a retrospective evaluation utilizing secondary data sources, researchers examine the results of a dementia care intervention that involved a clinical workflow, semiannual dementia training, and the integration of a care coordinator from an Area Agency on Aging (AAA) into the primary care team. Results: Seventeen caregivers received education and referrals to support during the intervention year and again in the final year. This represented 5.3% of the 322 patients diagnosed with dementia and with medical visits at the three clinics in 2023 During the last two grant years, there also was a large increase in provider referrals; thus, dementia care referrals decreased in proportion to patient referrals overall. Conclusions: Utilizing AAA care coordinators is a promising model for addressing health-related social needs in primary care. Yet, findings point to the complexities that remain in managing dementia in this setting.

## 1. Introduction

In the United States, dementia affects approximately 10% of people aged 65 years and older [1]. Despite the prevalence of this chronic disease, significant gaps persist in the care provided to individuals living with dementia [2,3]. A growing research base in primary care has documented complexities in supporting people living with dementia and their caregivers, including the lack of clinical workflows for dementia care with electronic health records (EHR) [4,5,6], fragmented or absent healthcare provider and clinic staff education [2,3,5,7,8,9,10], and missing interprofessional linkages and care management support [2,4,5,6,8,11,12,13]. In one response to these gaps, the Institute for Healthcare Improvement’s Age-Friendly framework focuses providers on evidence-based high-quality care that addresses the “4Ms”: What Matters, Medication, Mentation, and Mobility.

### 1.1. Clinical Workflows for Dementia Care

Primary care providers face challenges in delivering uniform or adequately tailored care to patients with dementia, particularly due to time constraints or the demands of managing other patients [2,3,9,14]. To address these challenges, the implementation of structured care models and/or workflows are identified as beneficial in the research [4,5,6]. Care for older adults can be streamlined through the implementation of workflows as well [5]. Workflows, such as EHR flow sheets and prompts integrated into patient charts, can be practical in highlighting key signs and symptoms that guide physicians toward an accurate diagnosis of dementia and help formulate appropriate action steps [4]. Ganz et al. (2008) found that structuring care models to support providers is valuable in under-resourced primary care settings that lack access to experts in the field [5]. Although evidence supports the effectiveness of implementing workflows and care models [4,5,6], replicability has been limited due to financial constraints, the need for provider training, and the time required for implementation [6].

### 1.2. Healthcare Provider and Clinic Staff Education

Staff education and training for primary care providers and clinic staff are key elements to optimize the care of people living with dementia [2,5,7,8,9]. Research highlights a clear need for comprehensive education in dementia care for primary care providers [2,7], including further training in conducting and evaluating cognitive screenings [10]. Likewise, Ganz et al.) highlight the need for increased education for the clinic staff in primary care to facilitate more comfortable interactions with vulnerable older adults [5]. Training clinical staff in addition to primary care providers is recommended [5,9]. However, although educational interventions may be somewhat effective in improving detection of dementia, these interventions do not appear to increase adherence to dementia guidelines [15]. Additional training needs include supporting patients with dementia in advanced care planning [9], addressing the mental health impacts of a dementia diagnosis, reducing the stigma surrounding brain change and dementia [10], and recognizing the benefits of early diagnosis [3].

### 1.3. Interprofessional Linkages and Care Management Support

Older adult health is uniquely complex, often involving interrelated and multifaceted needs [16] compounded by chronic conditions such as dementia [17]. A single-provider approach poses challenges in meeting these needs, whereas a multidisciplinary team provides enhanced benefits through more comprehensive care [4,5,6,8,11,12,13]. Multidisciplinary teams, which can include primary care providers, nurse practitioners, clinicians, and care managers, have been shown to significantly improve patient outcomes in dementia care programs, including fewer hospitalizations, reduced emergency department visits, and a reduction in long-term care admissions [8]. These effects are attributed to the collaborative care model, where the physician retains responsibility for the clinical aspects of care, while other members of the care team focus on addressing mental health, social well-being, and caregiver support [8]. Evidence demonstrates that a collaborative care model improves health outcomes for people living with dementia [4,5,8,11,12] including reducing symptom severity, enhancing quality of life, and reducing financial burden [12]. Similarly, the Chronic Care Model (CCM) emphasizes the positive impact of interdisciplinary teams and community resources in improving chronic disease management [18].

In particular, the involvement of a care manager or primary care liaison has been found to facilitate ongoing care by connecting patients to community organizations and essential resources such as Area Agencies on Aging, support groups, senior care, and other forms of assistance [8,11,19,20]. These connections enhance outcomes for both patients and caregivers by improving neuropsychological symptoms of dementia, reducing hospital admissions, and decreasing caregiver burden [11,19], while promoting cost-neutral care [8]. Care managers play a role in connecting patients to community organizations. These connections foster social engagement, peer connections, and physical activity, thereby supporting the emotional and physical well-being of individuals living with dementia [21]. Additionally, primary care provider referrals enhance the likelihood that people living with dementia and/or their caregivers will engage in post-appointment support, particularly, with a case manager [6,22]. Despite these findings, the integration of care managers into dementia care teams has been limited by the lack of case managers in primary care settings [20].

Our project attempted to address many barriers noted in the research literature. Most notably, the project introduced another discipline, care management from the Area Agencies on Aging (AAA), and a structured, EHR-based collaborative care model, into the primary care setting. In this role, the bachelor’s prepared AAA care coordinator was embedded in the clinic with access to the EHR and was available as a referral source for providers, linking patients and caregivers to education and support. The authors examined the change in the number of patients and caregivers who received education and support after the adoption of the dementia workflow in three primary care clinics. While provider utilization of the AAA care coordinator increased overall, AAA care coordinators remained underutilized with respect to caregiver dementia education and support.

## 2. Materials and Methods

Our mid-western public university was awarded a five-year Health Resources and Services Administration Geriatric Workforce Enhancement Program (GWEP) in 2019. A major objective was to transform primary care into age-friendly care by applying the Institute for Healthcare Improvement Age-Friendly Health System framework. Within the first fiscal year, the COVID-19 pandemic disrupted healthcare to such as extent that GWEPs had to shift focus to other objectives, such as the education of future health professionals, until they could return to the primary care setting. Our project did not implement the clinic-based intervention in dementia care until spring 2023. With the project concluding in a little more than a year later, the team planned a modest analysis of results across the three participating clinics, hoping to glean some implementation lessons and perhaps identify some promising data post-intervention.

The GWEP onboarded a primary care clinic in each of the first three years of the initiative. Working with the health system partner, the GWEP selected three clinics operating in medically underserved geographic areas. The GWEP also selected two clinics from different rural counties, one of which was a CMS-designated Rural Health Clinic (RHC). Table 1 provides demographic information about each clinical setting.

The urban clinic, Clinic A, was the first of the three to join the initiative. Clinic B, the larger of the two rural clinics, and the RHC, followed in the second grant year. The third clinic, Clinic C, represented a smaller, rural clinic. The dementia care intervention was implemented nine months after Clinic C formally joined the GWEP and the clinic’s AAA care coordinator had begun responding to provider referrals of patients.

### 2.1. Clinical Workflow Development and Implementation

To address mentation, an interdisciplinary team of staff working at the health system, AAA care coordinators, and university educators, with contributions from Alzheimer’s Associations partners, developed a clinical workflow to help ensure that patients diagnosed with dementia and their caregivers were receiving education and referrals to community resources. The clinical workflow was supported by step-by-step instructions for providers that populated the After Visit Summary with patient education on dementia and guided the generation of an EHR-based referral to the AAA care coordinator embedded in the clinic. Refer to Figure 1 for a graphical representation of the clinical dementia workflow implemented in GWEP primary care clinics. At each clinic, an Age-Friendly Champion provider was also identified to help promote provider awareness of age-friendly care practice changes, such as those outlined in Figure 1.

The clinical workflow was also supported by a detailed protocol for AAA care coordinators to follow when receiving a patient referral from a provider. The protocol guided AAA care coordinators to offer time for caregivers and patients to ask questions about the dementia diagnosis. It also walked care coordinators through state-funded caregiver programs and how to access additional education and support from the local chapter of the Alzheimer’s Association and the national Alzheimer’s Association Helpline.

The healthcare partner’s quality improvement staff trained providers, practice managers, and AAA care coordinators at each of the clinics on the new clinical workflow and accompanying EHR instructions. Key components shared with providers included information related to (a) talking to patients and caregivers about dementia diagnoses, (b) making EHR-based information about dementia available to patients and caregivers as a part of the After Visit Summary, and (c) referring the caregiver to the AAA care coordinator. In addition to providing one clinic-based workflow training for providers, the project also offered follow-up sessions with clinic practice managers and annual health system training on dementia care, delivered by the Alzheimer’s Association. During two grant years, the Alzheimer’s Association training was supplemented by medical content delivered by physicians practicing at the health system. Open to all health system employees, annual training was designed to raise general awareness about project goals in the three clinics and the resources available to the health system. Additionally, the healthcare partner’s quality improvement staff coached the AAA care coordinators to (a) provide information and support to caregivers in accessing and navigating local Alzheimer’s Association resources, (b) make a direct referral to the Alzheimer’s Association for free care consultation, and (c) distribute a free copy of Caring for A Person with Alzheimer’s Disease (National Institute on Aging, January 2019) to each caregiver.

### 2.2. Project Data Collection

In acknowledgement of the busy health system environment, and its substantial resources already committed to data collection, monitoring and reporting, the research team conducted a retrospective evaluation of the dementia care intervention. The evaluation utilized secondary data sources to examine whether dementia care interventions had impacted the primary care delivered to patients living with dementia and their caregivers. The university’s institutional review board determined that the project did not meet the definition of human subjects as set forth by Federal Regulations 45 CFR 46. A HIPAA Limited Data Set agreement was established and monitored by the health system’s research oversight and privacy board.

The retrospective evaluation utilized several secondary data sources from the health system to examine clinical outcomes. These sources are presented in Table 2 and are described in further detail below.

The health system’s quality improvement team extracted calendar-year EHR data for the period 2019–2024 to provide information on the volume of GWEP clinic patient populations and annual clinical visits. Utilizing CMS specifications, the quality improvement team also used EHR data to calculate the denominator of MIPS 288. The MIPS 288 denominator offered a standardized and validated measurement of the patient population living with dementia across GWEP clinics, ensuring that the same exclusionary and inclusionary criteria were used [23]. Because the quality improvement team already had a significant CMS MIPS reporting program, they found this role a relatively light burden. Given that data represented aggregated counts, the research team did not have access to other related characteristics of patients living with dementia, such as the date of dementia diagnosis or an identified healthcare representative in the medical chart.

AAA care coordinator service records were also gathered. These provided information on the total number of patients referred to care coordinators each year, the number of patients and/or caregivers who received some form of assistance from a care coordinator, and the number of caregivers who received dementia-specific education and support referrals. AAA care coordinators were asked to double-enter records of patient services, once in a standardized spreadsheet, followed by a provider note in the patient medical chart. Appendix A defines the fields contained within the care coordinator tracking spreadsheet. Appendix A presents the referral options that were available to select and the referral type that was associated with that option. As shown in Appendix A, care coordinators recorded each patient interaction, noting whether the interaction reflected a new provider referral or was a follow-up contact with a previously referred patient and what assistance was provided during the interaction, such as assistance with food stamps or a Medicaid application or a referral to the Meals on Wheels program or the Alzheimer’s Association. See Appendix A for a complete list of the variables contained in the spreadsheet.

Care coordinators stripped patient identifiers such as name and Medical Record Number before uploading encrypted spreadsheets to a secure cloud-based file directory. A HIPAA Limited Data Set agreement allowed the data set to retain birth year, residential ZIP code, and date of care coordinator contact attempt. Only patient-level data stripped of patient identifiers were available from care coordinators; therefore, the research team was not able to link patient data across different care coordinator interactions for statistical analysis.

Data sources presented another challenge, even to descriptive analysis. MIPS methodology tied data measurement to the health system’s annual measurement schedule based on the calendar year. In contrast, care coordinator data were reported on the grant year cycle, 1 July through 30 June, reflecting the annual cycle of GWEP programmatic activities and the onboarding of clinics. Table 3 shows how the measurement period of data sources differed, depending on whether the source was the AAA care coordinator services form or MIPS measurement.

Because MIPS data on the number of patients living with dementia straddled two grant years, it was not possible to know the precise number of patients living with dementia and their caregivers who were not served each year. Care coordinator data generally matched the grant year, except for the first year each clinic joined due to the time needed to train AAA care coordinators in the EHR and establish referral processes. Additionally, the COVID-19 pandemic impacted early referral work at Clinic A and appeared to contribute to a very slow start-up of care coordinator services at Clinic B during the second grant year. Potential analyses were hamstrung by available data sources and the sustained effects of COVID-19 on healthcare delivery systems during the initiative.

## 3. Results

To understand clinical settings of the GWEP dementia intervention, the research team examined data on the patient population of older adults (aged 65 and older) served by each clinic. As Table 4, below, shows, the older adult population made up a significant proportion of each clinic’s total patient population. Even so, the proportion of annual clinic visits made by older adults was even larger than that of the population: older adults at Clinic A made up 44% of the patient population and nearly 52% of annual clinic visits; similarly, older adults made up 29% and 27%, respectively, of Clinic B and Clinic C patient populations and 35% of annual clinical visits.

The sizeable older adult patient populations and clinic locations in medically underserved areas made these clinics good candidates for GWEP interventions, designed to improve the quality of healthcare of older adults.

Next, the research team wanted to understand more about the prevalence of dementia at participating clinics. Table 5 shows the number of patients diagnosed with dementia and with at least one clinical visit during 2023. Because these data were derived from the denominator of the dementia MIPS measure, this number reflects all patients with dementia, not just those aged 65 and older.

Across the three clinics, there were 304 patients with a medical diagnosis of dementia. Because this number includes patients aged 64 and younger with a medical dementia diagnosis, the percentage of each clinic’s 65-and-older population was less than the figure estimates.

Comparing the expected number of patients living with dementia at each clinic, the data suggested that the patient population with a medical diagnosis of dementia was likely an underrepresentation of the actual patient population living with dementia. At Clinic A and B, patient populations diagnosed with dementia represented approximately half of what would be expected based on national estimates for patients aged 65 and older, alone.

The number of caregivers who received education and referrals to community support was even smaller than the diagnosed population, as reflected in Table 6. During the 5 years of the GWEP initiative, AAA care coordinators provided dementia education and referrals to support for 88 caregivers of patients living with dementia. Breaking these unduplicated referral numbers down by clinic showed that far fewer than half of diagnosed patients were assisted by AAA care coordinators.

To assess whether there had been an increase in patient referrals following the implementation of the dementia intervention at clinics, the research team disaggregated referral data by year at each clinic (Table 7, Table 8 and Table 9).

The data in Table 7 indicated that each year, patients living with dementia made up a small fraction of the patients that providers referred to AAA care coordinators. Fourty-two caregivers, or 65.6% of those assisted, received assistance prior to the intervention year. Another 22 caregivers received assistance during the intervention year and the final year of the initiative. During this time, the share of patient referrals made up by medically diagnosed dementia patients declined.

Similar patterns were observed in Clinics B and C. At Clinic B, the only provider referrals of patients with dementia came before the dementia clinical workflow was introduced.

The dementia care intervention did not increase the referral of patients diagnosed with dementia at Clinic C either.

Altogether, 54 of the 88 caregivers who received assistance during the GWEP received it before the intervention was implemented, with the remaining 34 in the last two years of the initiative. Seventeen caregivers received education and referrals to support during the intervention year and again in the final year. During the same period, provider referrals of patients increased, translating into a decrease in the percentage of patients diagnosed with dementia represented in annual patient referrals.

## 4. Discussion

Results from this study suggested that the intervention did not take hold as intended. After the introduction of the clinical workflow for patients diagnosed with dementia, clinic providers did not take advantage of most clinical encounters with these patients to make a referral to the AAA care coordinator. Yet, 1714 patients received referrals to the AAA care coordinator and received assistance accessing healthcare and other health-related social needs during the five-year initiative. Furthermore, 450 or 26.25% of referred patients were connected to an AAA care coordinator for assessment and program eligibility for a range of services including home care and home health. This meant additional opportunities for caregiver engagement and referrals to support as patients entered the agency’s service population. Thus, patient connections to the AAA agency would offer another opportunity to identify cognitive challenges and caregiver needs.

Across the three clinics, the number of patients referred to AAA care coordinators by providers increased substantially in the year that the dementia workflow was introduced. This increase may have reflected a new awareness of providers regarding the vulnerabilities of older adults, following COVID-19. However, this increase did not convert to more patients and caregivers receiving dementia education and referrals to support. The number of patients with a dementia diagnosis and a medical encounter at either of the three clinics who were not referred to the care coordinator is striking. A gap between the number of patients living with dementia seen each year and those provided with a referral persisted during the project. This gap could be partially attributed to the fact that dementia symptoms are broad and go beyond memory loss to include multiple types of cognitive changes such as communication deficits, cognitive processing, difficulty with activities of daily living and instrumental activities of daily living.

The project team anticipated that the integration of the AAA care coordinator into the dementia clinical workflow would promote its adoption and eventual institutionalization, consistent with the underpinnings of the CCM and its emphasis on linkages between the health and social needs of patients [5]. Although dementia workflow training was delivered to providers and office staff in three primary care clinic sites, the data underscore the limitations of one-time or infrequent training of providers, without more rigorous intervention to support clinical practice as indicated in the literature [5,9]. This finding aligns with Perry et al., who highlighted that the length and frequency of educational interventions affected the depth of understanding that family practitioners gained regarding dementia identification [15]. Additionally, these data suggest the need for ongoing provider education regarding the benefits of early detection as well as continued reinforcement of the value-add in utilizing the case manager to support people living with dementia and their caregivers in primary care [11].

Finally, it is worth noting that the stigma associated with dementia and reluctance on the part of the provider to screen [2], refer the patient to a neurologist for diagnosis, or offer support in managing a diagnosis of dementia may have played a factor in the underrepresented number of provider referrals specific to dementia. Project results appear to suggest that a more comprehensive intervention is needed, one that addresses both patient and provider concerns about a potentially stigmatizing diagnosis.

Results from this descriptive study point to the complexities that exist in managing dementia in primary care in two areas: (a) the gap between patients diagnosed with dementia and their caregivers and those who were provided education for support; and (b) the need for ongoing provider education in dementia care practice, which is consistent in the literature [5,9]. Additionally, the research raises questions about the role that stigma plays in dementia diagnosis as well as the role of competing provider demands. The schematic diagram in Figure 2 highlights the gap referenced above and lists methods to reduce this gap to include a) increased provider and clinic staff education about dementia, (b) reduced stigma about dementia among patients, providers, caregivers, and staff, (c) embedded AAA care coordinators in primary care, and (d) the promotion of electronic health record dementia workflow adoption.

Despite the limited number of patients and caregivers benefiting from the dementia intervention, this study adds to the health and human service integration literature [24]. Primary care providers utilized the embedded AAA care coordinator to improve patient care, demonstrating the feasibility of addressing health-related social needs in the primary care clinical setting and supports. During the initiative, 1714 patients from the three clinics were assisted in advance care planning, referrals to evidence-based health programs, and referrals or assistance accessing community-based resources. In this context, the CCM becomes an important framework, as it encourages interdisciplinary collaboration to direct care improvement for chronic conditions [18], like dementia. This model’s focus on integrating quality care with community support aligns with the integration of the AAA care coordinators in primary care settings.

## 5. Conclusions

This modest research project helps illuminate some of the complexities that exist in managing dementia care in primary care. Uncovering complexities in managing dementia is challenging in health services research where there is typically a wealth of existing data, which makes primary data collection a difficult case to make to administrators. Because of this, significant technical expertise and health system partnership is needed to identify the measures approximating study variables. Furthermore, the sheer volume of data requires significant technical expertise to access and manipulate for new variable construction and aggregation.

Despite its modest scope, the study also suggests that Area Agencies on Aging and primary care clinics can develop partnerships that result in better primary care for the aging population. The research team looks forward to additional development and testing of partnership models between AAAs and primary care, as the healthcare field continues to explore interventions that integrate medical and social care.

Finally, the project outcomes lend insight into implications for clinical practice, state health policy, and future research. The project raises some important questions about efforts to make primary care best practice the easy choice in EHR workflows. The research team found that while workflows can help integrate healthcare delivery by multidisciplinary teams, workflow development and adoption, in and of itself, is an insufficient strategy for improving dementia care. Additional strategies are needed. These might include state mandated dementia training hours for annual license requirements for providers. Other future strategies might embed artificial intelligence in medical software to prompt dementia smart phrases, referrals, and educational handouts. Creating certificate programs and/or mini fellowships for primary care providers who have a strong interest in eldercare also may help with the hesitancy to address dementia. Designation of one to two of these specially trained providers in a group practice may enhance care for elders in rural or urban communities.

Moving forward, the GWEP project will continue to embed AAA care coordinators in primary care clinics and support this integration through EHR-based workflows to connect patients and caregivers to valuable community resources and to prompt providers in decision-making. However, a Geriatric Provider Support Model will be implemented to support this work, making training, not the EHR workflow, the foundation of new care practices. Clinic adoption and implementation of the model will be aided by dedicated mentoring and support roles of a nurse practitioner and medical office assistant. In addition, more attention will be given to the national efforts of the Alzheimer’s Association in defining dementia workflows for primary care.

## 6. Study Limitations and Contributions

Given the many limitations of the data, and the challenges of implementing any intervention in the post-COVID-19 period, statistical analyses comparing the time periods before and after intervention implementation were neither feasible nor appropriate. However, the research team still believed there was value in contextualizing its work and examining patterns descriptively, given growing research interest in effective dementia care intervention. Positive outcomes were limited to the relatively few patients and caregivers who received assistance and appeared to be attributable to the AAA care coordinators embedded in primary care clinics, not other aspects of the intervention, namely, the clinical workflow, which had promised to guide providers in referring patients diagnosed with dementia and their caregivers to education and community resources. Barriers to accessing dementia education and support remained.

Despite limitations, the study highlights some of the implicit challenges of introducing dementia care interventions, even when the burden to clinical teams is at the forefront of the intervention’s design. This intervention targeted only patients diagnosed with dementia; as research literature has suggested, this is likely only a portion of the patient population needing these resources. Furthermore, the intervention required providers to make a patient referral; this likely involved an explanation for the referral and the confronting of the diagnosis by both provider and patient. The dementia care intervention at the center of this study had a modest objective, brief education of caregivers and assistance in accessing community resources available to support the patient and the caregiver in living with a chronic disease. The barriers to this project indicate that more systemic health promotion strategies are required to deliver better dementia care.

## Figures and Tables

**Figure 1 ijerph-22-00506-f001:**
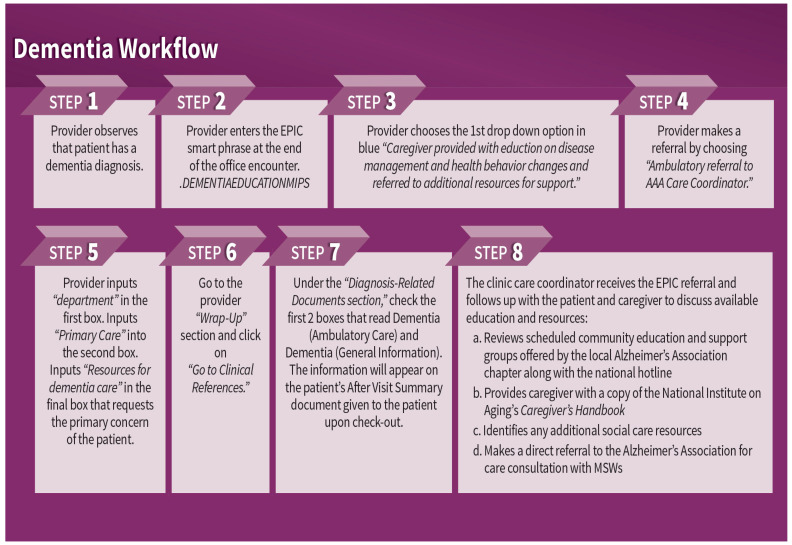
Age-friendly dementia workflow for interdisciplinary clinical team.

**Figure 2 ijerph-22-00506-f002:**
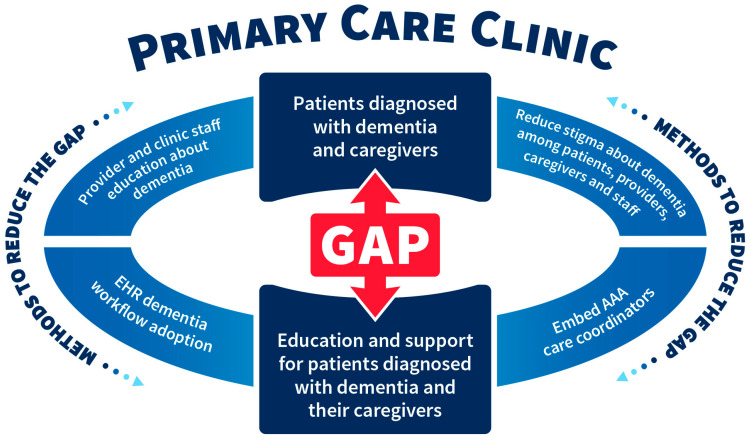
Existing gaps in dementia care within primary care clinics and methods to address these gaps.

**Table 1 ijerph-22-00506-t001:** GWEP clinics by setting descriptors.

Clinic Case Identifier	Number of Annual Clinic Visits (2023)	Medical Service Area Designation	Primary Care Health Professional Shortage Score	Population Density Category	Number of Providers
A	21,193	Medically Underserved Area (Census Blocks)	16	Urban	8.5 *
B	15,591	Medically Underserved Area (County)	10	Rural	5 **
C	9831	Medically Underserved Area (County)	11	Rural	3 ***

* There are nine providers at the clinic with one serving as both an obstetrician and a family medicine physician. The dual role of this physician is why only 8.5 providers were counted. Note that not all providers were physicians. At this clinic, one provider was a nurse practitioner. ** Two of the five providers were nurse practitioners. *** One of the three providers was a nurse practitioner.

**Table 2 ijerph-22-00506-t002:** Secondary data sources used to examine intervention promise.

Data Source	Data Element(s)	Rationale
EHR Data	Annual patient population; annual Medicare population; annual clinic visits; (January-December annually)	Describe clinical settings
MIPS Clinical Quality Measure 288, Dementia	CMS-specified Measure 288, Dementia: Caregiver Education and Referral to Support (January-December annually)	Provide an unduplicated count of target population
Area Agency on Aging Care Coordinator	patient services data Clinic A (19 October–24 June) * Clinic B (20 August–24 June) Clinic C (21 October–24 June)	Describe intervention population and obtain service counts

* Grant budget-years started from 1 July 2019–2023.

**Table 3 ijerph-22-00506-t003:** Dementia data measurement schedule.

Initiative Grant Cycle	AAA Care Coordinator Service Data	MIPS Denominator Measurement
GWEP Year 1 (July 2019–June 2020)	Clinic A (October 2019–June 2020)	January–December 2019
GWEP Year 2 (July 2020–June 2021)	Clinic A Clinic B (August 2020–June 2021)	January–December 2020
GWEP Year 3 (July 2021–June 2022)	Clinics A and B Clinic C (September 2021–June 2022)	January–December 2021
GWEP Year 4 (July 2022–June 2023)	Clinics A–C	January–December 2022
GWEP Year 5 (July 2023–June 2024)	Clinics A–C	January–December 2023

**Table 4 ijerph-22-00506-t004:** GWEP clinic demographics, 2023.

Clinic Case Identifier	Patient Population	65+ Patient Population	%	Annual Clinical Visits	Clinical Visits 65+ Patients Only	%
A	7984	3546	44%	21,193	10,976	52%
B	5203	1499	29%	15,591	5508	35%
C	2859	768	27%	9831	3394	35%

**Table 5 ijerph-22-00506-t005:** Number of patients diagnosed with dementia versus the expected number, 2023.

Clinic	Number of Patients Diagnosed with Dementia	% of 65 and Older Patient Population	Number Expected	%
A	173	<5%	355	10%
B	61	<4%	150	10%
C	70	<9%	77	10%
Total	304		582	

Note: The expected number reflects the 10% prevalence rate of Alzheimer’s Disease and other dementias among people aged 65 and older [1].

**Table 6 ijerph-22-00506-t006:** Total number of caregivers receiving education and support at GWEP clinics.

Clinic	Number of Caregivers Educated and Referred to Support by AAA Care Coordinators	% of Diagnosed Patients
A	64	33%
B	5	8%
C	19	27%
Total	88	

Note. Caregivers who were not reached or who declined assistance were not included in the count of caregivers educated and referred to support by AAA care coordinators.

**Table 7 ijerph-22-00506-t007:** % of provider referrals resulting in a dementia care intervention, Clinic A.

Participation Year	Total Number of Provider Referrals	Number of Caregivers Receiving Dementia Education/Support	% of Provider Referrals Resulting in Dementia Care
GWEP Year 1 (July 2019–June 2020)	82	6	7%
GWEP Year 2 (July 2020–June 2021)	66	18	27.3%
GWEP Year 3 (July 2021–June 2022)	82	18	22%
**GWEP Year 4** **(July 2022–June 2023)** **Intervention Year**	**105**	**9**	**9%**
GWEP Year 5 (July 2023–June 2024)	113	13	12%
Total	448	64	

Note. Bold indicates intervention year. Caregivers who were not reached or who declined assistance were not included in the count of caregivers educated and referred to support by AAA care coordinators.

**Table 8 ijerph-22-00506-t008:** % of provider referrals resulting in dementia care intervention, Clinic B.

Participation Year	Total Number of Provider Referrals	Number of Caregivers Receiving Dementia Education/Support	% of Provider Referrals Resulting in Dementia Care
GWEP Year 2 (July 2020–June 2021)	60	3	5%
GWEP Year 3 (July 2021–June 2022)	127	2	2%
**GWEP Year 4** **(July 2022–June 2023)** **Intervention Year**	**319**	**0**	**0%**
GWEP Year 5 (July 2023–June 2024)	404	0	0%
Total	910	5	

Note. Bold indicates intervention year. Caregivers who were not reached or who declined assistance were not included in the count of caregivers educated and referred to support by AAA care coordinators.

**Table 9 ijerph-22-00506-t009:** % of provider referrals resulting in dementia care intervention, Clinic C.

Participation Year	Total Number of Provider Referrals	Number of Caregivers Receiving Dementia Education/Support	% of Provider Referrals Resulting in Dementia Care
GWEP Year 3 (July 2021–June 2022)	29	7	24%
**GWEP Year 4** **(July 2022–June 2023)** **Intervention Year**	**185**	**8**	**4%**
GWEP Year 5 (July 2023–June 2024)	142	4	3%
Total	356	19	

Note. Bold indicates intervention year. Caregivers who were not reached or who declined assistance were not included in the count of caregivers educated and referred to support by AAA care coordinators.

## Data Availability

The datasets presented in this article contain private health information and cannot be made publicly available. Data inquiries should be directed to Mary C. Ehlman.

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
