# Peer review of "Integrating AAA Care Coordinators in Primary Care for Dementia Support: Implementation Challenges and Lessons Learned"

_ijerph, 2025, doi:10.3390/ijerph22040506_

Round 1
Reviewer 1 Report
Comments and Suggestions for Authors
Authors in the current manuscript explored implementing EHR-based workflows and embedding care managers to improve dementia care in primary clinics but found limited impact due to under-utilization of resources and gaps in provider training. It recommends ongoing education, addressing stigma, and fostering interdisciplinary collaboration to enhance outcomes. Overall manuscript is straightforward story should be accepted as it is provided it fits into the theme of the journal.
I have two additional suggestions for authors:
i) Please add a few sentences mentioning conlusions/findings in abstract. Current abstract is quite unclear in what readers should expect.
ii) Please mention any ethical approvals that were used for these studies.
iii) Please add a schematic diagram in the end that should clearly reflect on suggestions made in the manuscript.
Author Response
|
Reviewer 1 - Thank you for your helpful comments. We appreciate your time, expertise and suggestions. Below is a list of your comments and our responses/changes. |
||
|
# |
Comment |
Changes |
|
1A |
Adjust abstract to include any major findings and conclusions. |
Added sentence to reflect major findings and conclusions to abstract. See 2 D below for more on abstract adjustment to include major findings. |
|
1B |
Address any ethical approvals used for the study. |
Added sentences to 2.2 Project Data Collection and addressed HIPAA LDS. Also, strengthened IRB Statement. |
|
1C |
Schematic Diagram reflecting suggestions made*** |
Schematic diagram and explanation added to the Discussion |
Reviewer 2 Report
Comments and Suggestions for Authors
COMPREHENSIVE MANUSCRIPT EVALUATION REPORT
I. General Considerations
Strengths:
- The study addresses a highly relevant and timely topic in the fields of public health and geriatrics.
- The overall methodology demonstrates alignment with the study's objectives.
- The manuscript exhibits a logical and coherent structure.
- The presented tables and data are informative and well-organized.
- The study effectively integrates critical concepts such as primary care, dementia management, and caregiver support.
Weaknesses:
- Certain sections of the manuscript would benefit from more in-depth analysis.
- The integration between the results and existing literature could be more robust in some areas.
- Some study limitations are not addressed with sufficient detail.
Suggestions for Improvement:
- Enhance the discussion of results, particularly in relation to extant literature.
- Include a more comprehensive section on study limitations.
- Improve the integration of supplementary files with the main text.
II. Section-Specific Considerations
- Title
Strengths:
- The title "Addressing Dementia through New Primary Care Workflows: Implementation, Challenges, and Implications" is clear and informative.
- It effectively captures the key elements of the study: dementia, primary care, novel workflows, implementation, and challenges.
Weaknesses:
- The title does not specifically mention the integration of Area Agency on Aging (AAA) care managers, which is a unique aspect of the study.
Suggestions for Improvement:
- Consider incorporating a reference to the integration of AAA care managers, for example:
- "Challenges in Implementing Dementia Care Workflows in Primary Care: Insights from AAA Care Manager Integration in Midwest U.S. Clinics"
- "Limited Adoption of New Dementia Care Workflows in Primary Care: A Study of AAA Care Manager Integration and Caregiver Support"
- "Barriers to Enhancing Dementia Care in Primary Settings: Evaluation of a Workflow Intervention with AAA Care Managers"
- "Integrating AAA Care Managers in Primary Care for Dementia: Implementation Challenges and Lessons Learned"
- "Bridging Primary Care and Community Resources for Dementia: Outcomes of a Workflow Intervention with AAA Care Managers"
- Abstract
Strengths:
- Provides a concise overview of the study, including context, objectives, methods, and key findings.
- Mentions the integration of AAA care managers, which is a unique aspect of the study.
Weaknesses:
- Lacks specific numerical data or percentages for key findings.
- Does not explicitly state implications or recommendations based on the results.
Suggestions for Improvement:
- Include key quantitative data, such as the total number of patients referred and the percentage who received dementia education.
- Add a sentence on the implications of the findings or recommendations for future interventions.
- Consider structuring the abstract with clear subsections (Background, Objectives, Methods, Results, Conclusions) to enhance clarity and facilitate rapid reading.
- Introduction and Theoretical Framework
Strengths:
- Establishes a solid theoretical foundation for the study.
- Cites relevant and up-to-date literature on challenges in dementia management in primary care.
- Addresses important concepts such as clinical workflows, provider education, and caregiver support.
Weaknesses:
- Could provide a more detailed contextualization of the specific problem addressed by the study.
Suggestions for Improvement:
- Expand the discussion on similar interventions conducted in other contexts.
- Strengthen the rationale for the specific approach chosen in this study.
- Methodology
Strengths:
- Clearly describes the development and implementation of the clinical workflow.
- Provides details on data collection and sources utilized.
Weaknesses:
- Lacks details on potential methodological limitations.
Suggestions for Improvement:
- Include a more detailed discussion on the potential limitations of the chosen methodology.
- Elucidate more clearly how data were analyzed and interpreted.
- Results
Strengths:
- Presents clear and objective data through well-structured tables.
- Provides a comprehensive view of results across different clinics and over time.
Weaknesses:
- Lack of more robust statistical analyses to assess the significance of observed changes.
Suggestions for Improvement:
- Include formal statistical analyses to evaluate the significance of changes observed after workflow implementation.
- Provide a more detailed analysis of trends over time and an explicit comparison between pre- and post-intervention periods.
- Discussion
Strengths:
- Relates the findings to existing literature.
- Acknowledges the discrepancy between the number of patients diagnosed with dementia and those who received education and support.
Weaknesses:
- The discussion could delve deeper into the analysis of specific results.
- Lacks a more systematic comparison with similar studies.
Suggestions for Improvement:
- Deepen the analysis of how the results align with or challenge existing theoretical models of dementia management in primary care.
- Include a section that systematically compares the obtained results with similar studies, highlighting similarities and differences.
- Strengthen the explanation of discrepancies between the obtained results and expectations based on literature.
- Conclusions
Strengths:
- Address the main objective of the study.
- Acknowledge that the intervention did not have the expected impact.
Weaknesses:
- Some conclusions could be more specific and directly linked to the presented data.
- Lacks a clear and concise synthesis of the main findings of the study.
Suggestions for Improvement:
- Begin the conclusions section with a clear and concise synthesis of the study's main findings.
- Elaborate more thoroughly on the specific implications of the results for clinical practice, health policies, and future research.
- Provide more specific recommendations for future interventions and research.
- Contributions and Limitations
Strengths:
- Briefly acknowledges some study limitations.
Weaknesses:
- There is no specific section dedicated to the study's contributions and limitations.
- The mentioned limitations are too general.
Suggestions for Improvement:
- Add a dedicated section at the end of the manuscript titled "Contributions and Limitations of the Study."
- Explicitly list the main contributions of the study.
- Detail specific study limitations, such as sample size, potential biases, limitations in generalizing results, and implementation challenges.
- Discuss how the identified limitations may have impacted the results and conclusions of the study.
- References
Strengths:
- The references are pertinent and up-to-date, with 61.9% published in the last five years.
- Adequately cover the main aspects of the investigated topic.
Weaknesses:
- Could include more comparative studies on similar interventions.
Suggestions for Improvement:
- Consider including more comparative studies that have implemented similar interventions in primary care for dementia.
- Potentially expand references related to specific barriers in implementing new workflows in primary care.
- Ethical Aspects
Strengths:
- Explicitly mentions approval from the Institutional Review Board.
- Demonstrates compliance with federal regulations.
Weaknesses:
- No significant weaknesses identified in this section.
Suggestions for Improvement:
- No significant suggestions for this section.
- Supplementary Files
Strengths:
- Provide detailed and relevant information on data collection and referral categorization.
- The tables are clear and easy to understand.
Weaknesses:
- The formatting of the tables could be improved for greater readability.
- Lack explanatory notes for some terms or abbreviations used.
Suggestions for Improvement:
- Add a brief introduction or legend for each table.
- Include footnotes to explain abbreviations or technical terms.
- Improve table formatting, possibly using grid lines or alternate shading.
- In the main text, make more specific references to these tables when discussing data collection and referral analysis.
- Consider adding a column in Table S2 indicating the frequency of each type of referral.
- Check consistency between terms used in supplementary tables and the main text of the manuscript.
This comprehensive report provides a detailed view of the strengths, weaknesses, and areas for potential improvement throughout the manuscript. The offered suggestions aim to strengthen the work in various aspects, from data presentation to the depth of analysis and discussion. By addressing these points, the authors can enhance the overall quality, clarity, and impact of their research contribution.
Author Response
|
Reviewer 2 - Thank you for your helpful comments. We appreciate your time, expertise and suggestions. Below is a list of your comments and our responses/changes. |
||
|
2A |
Enhance the discussion of the results in relation to current literature. |
Added at least 4 references of current literature to the discussion |
|
Title |
||
|
2B |
Adjust the title to incorporate the role of the AAA managers. |
Title edited to incorporate care manager/care coordinator |
|
Abstract |
||
|
2C |
Address the findings and conclusions in the abstract. |
Addressed in 1A |
|
2D |
Include quantitative data within the abstract. 250 words |
See new results section recommended in 2E. The results include quantitative data from major findings in Results section as suggested by this reviewer and reviewer 1. |
|
2E |
Arrange the abstract with clear subsections to enhance clarity. (background, materials and methods, etc.) |
Adjusted the abstract to include background, methods, results, conclusions |
|
Introduction |
||
|
2F |
Expand on the rationale behind the approach. |
Added explanation in section 1.3 including using the Chronic Care Model |
|
2G |
Expand on similar interventions conducted in other contexts. |
Added Age-Friendly 4Ms, Chronic Care Model, and; and citation 21 and 22. |
|
Methodology |
||
|
2H |
Provide more detail on methodological limitations. |
Added more detail on methodological limitations throughout the Materials and Methods section |
|
2I |
Elucidate on the method for data analyzation and interpretation. |
Due to the methodological limitations discussed, the analysis is descriptive. More detail about the descriptive analysis method is provided. |
|
Results |
||
|
2J |
Include more formal statistical analyses to evaluate the change after workflow implementation. |
Formal statistical analyses would suggest a stronger research design, and no statistical analysis is needed to observe the increase of provider referrals and the continued low rate of dementia caregivers receiving education and referrals to support. |
|
2K |
Provide a more detailed analysis of the trends over time and comparison of pre and post intervention. |
More detail about patterns over time was provided in the results section. |
|
Discussion |
||
|
2L |
Dive deeper into the analysis of specific results elaborating on how they challenge existing theoretical models of dementia management in primary care. |
Connected findings to the Chronic Care Model (CCM) |
|
2M |
Compare the obtained results with similar studies, highlighting their similarities and differences. |
Added connections with additional literature in the discussion section |
|
2N |
Strengthen explanation of discrepancies in the obtained vs expected results. |
Added a column in table in Results section; Moved paragraph from results to discussion; Addressed gap identified |
|
Conclusion |
||
|
2O |
Begin with a more clear and concise synthesis of the main findings. |
Edited first and second sentence of the conclusion. |
|
2P |
Elaborate more thoroughly on the specific implications of the results for clinical practice, health policies and future research. |
Expanded the third paragraph of the conclusion to include specific implications of the results for clinical practice, health policies and future research. |
|
2Q |
Provide more specific recommendations for future interventions and research |
Expanded the third paragraph of the conclusion to include specific recommendations for future interventions and research
|
|
NEW section discussing Limitations and Contributions |
||
|
2R |
Explicitly list the main contributions of the study. |
Addressed in the new Limitations and Contributions section that the reviewer recommended. |
|
2S |
Detail the study limitations such as sample size, biases, limitations in generalizing results, and implementation challenges. |
Detailed study limitations are now addressed in the Materials and Methods section. They are revisited in the new Limitations and Contributions section that the reviewer recommended. |
|
2T |
Discuss how the limitations impact the results. |
Addressed in Limitations and Contributions section that the reviewer recommended.
|
|
References |
||
|
2U |
Consider citing more comparative studies. |
Refer to 2M reviewer comment |
|
2V |
Expand references to include specific barriers in implementing new workflows in primary care. |
Added article from Menn citing physician referral as a reason for care manager use. |
|
Ethical Implications |
||
|
2W |
No changes need to be made in this field. |
Contradicts comment made in 1B |
|
Supplementary Files |
||
|
2X |
Add a brief introduction or legend for each table. |
Brief description of the contents of the table added. |
|
2Y |
Include footnotes to explain abbreviations and terminology. |
Added a footnote to the table to explain abbreviations. |
|
2Z |
Improve formatting, incorporating grid lines and shading. |
Added shading to improve readability and just formatting overall. |
|
2AA |
Make more specific connections to the supplementary materials in the main text. (When discussing data collection and referral analysis.) |
Supplementary tables are now mentioned on page 6, where AAA care coordinator service tracking sheets are described, and on page 9 in the Results section |
|
2BB |
Add column in Table S2 indicating frequency of each type of referral. |
The purpose of the S2 Table was to show the types of referral options that AAA care coordinators had available and how these were coded analytically to the 4Ms and major categories of health-related social needs (e.g., food assistance). Presenting frequency data would suggest that one could reliably compare utilization across options. Many of these categories changed over time and in some cases did not apply to all three clinics. For example, ACP data were not collected at Clinic A due to a different workflow implemented there that engaged an ACP facilitator and not the care coordinator. |
|
2CC |
Check consistency between terms used in supplementary and main texts. |
Writing team checked for inconsistencies. |
|
1A |
Adjust abstract to include any major findings and conclusions. |
Added sentence to reflect major findings and conclusions to abstract. See 2 D below for more on abstract adjustment to include major findings. |
|
1B |
Address any ethical approvals used for the study. |
Added sentences to 2.2 Project Data Collection and addressed HIPAA LDS. Also, strengthened IRB Statement. |
Reviewer 3 Report
Comments and Suggestions for Authors
The authors have done an excellent job addressing an important and meaningful topic. The study is well-written, and the subject matter holds significant relevance, particularly in the context of community-based care for individuals with dementia. However, there are some points that might be useful.
In the first paragraph, the author mentioned the prevalence of dementia in the USA. As the present study is conducted in southern India, it would be beneficial to include some data on the prevalence of dementia in the authors' country.
On Page 2, line 98, regarding the care manager of AAA, could the authors provide more details about their background? Are they professionals with appropriate qualifications? On Page 4, line 137, the authors mentioned a training program. Could the authors elaborate on its general content, as well as its duration, frequency, and other relevant details?
The present study is descriptive research. Based on the authors' data, it seems that the number of people with dementia transferred to the community is very low. Analyzing the potential reasons behind this trend might be valuable.
Some data mentioned the support from caregivers, including the proportion willing to receive education. It appears that the proportion of caregivers willing to provide assistance is higher in Clinic A compared to Clinics B and C. This is noteworthy, as Clinics B and C are rural hospitals, where caregivers may face a heavier burden.
Author Response
|
Reviewer 3 - Thank you for your helpful comments. We appreciate your time, expertise and suggestions. Below is a list of your comments and our responses/changes.
|
||
|
3A |
Adjust the first source to assess the prevalence of dementia from the United States. |
Replaced with CDC study. |
|
3B |
Pg 2 Line 98, Provide more details on the background of the AAA manager. Are they professionals with appropriate qualifications? |
Added bachelor’s prepared from AAA job description |
|
3C |
Pg 4, Line 137, Elaborate on the general content of the training program, duration, frequency and other relevant details. |
Added detail to key components of training for providers and AAA care coordinators. Included follow-up with practice managers. |
|
3D |
Analyze potential limitations behind the limited impact. |
Addressed in comment 2T |
|
3E |
Note the discrepancy of caregivers willing to receive education between A, b, c. This may be due to B and C being rural hospitals, where caregivers may face a higher burden. |
Noted in results; Mentioned rural/urban in conclusion related to provider training |
|
2T |
Discuss how the limitations impact the results. |
Addressed in Limitations and Contributions section that the reviewer recommended.
|
Reviewer 4 Report
Comments and Suggestions for Authors
The study addresses the complexities of managing dementia in primary care and the need to improve patient care and family support, as well as the training needs of those involved in caring for patients with dementia in primary care. Ideas are provided for interesting new models and workflows to develop in primary care, although the sample size in the study is small, a limitation explicitly acknowledged by the study authors.
Author Response
|
Reviewer 4 - Thank you for your helpful comments. We appreciate your time, expertise
|
||
|
4A |
No Comments Made |
|
Reviewer 5 Report
Comments and Suggestions for Authors
Thank you for the opportunity to review this manuscript. The manuscript entitled “Addressing Dementia through New Primary Care workflows: Implementation, challenges, and implications” examines the implementation of a primary care intervention that hopes to increase education and support to patients and family members with dementia through the addition of a AAA care. Although the results of this study did not demonstrate successful integration of this program it sheds light on an important topic and a gap in knowledge and intervention around dementia care. Below are comments I believe will strengthen your paper.
Introduction/Materials and Methods:
- I found your introduction to be thorough however I do think it would be useful to include a description of the 4MS model in the introduction rather than the materials/methods as this appears to be important to why you were doing this study. I think it can be fairly brief and focus mainly on the mentation component but may be useful for readers who are less familiar with this framework.
Discussion/Conclusions
- Though the results of your study are disappointing it does shed light on a large issue in Primary care settings and raises questions of why is there such a gap between having a dementia diagnosis and refer to care. I think your paper could improve with a more robust discussion section that provides further hypothesis of why there was such a gap- perhaps its that dementia was too broad and providers did not feel that the person had severe enough dementia to warrant a referral or perhaps as you mentioned, they felt uncomfortable making the referral. I am also curious if this notes people who maybe were provided information at the appointment but declined a referral?
Author Response
|
Reviewer 5 - Thank you for your helpful comments. We appreciate your time, expertise and suggestions. Below is a list of your comments and our responses/changes.
|
||
|
5A |
Provide a brief description of the 4M’s in the introduction rather than the materials and methods, highlighting the mentation component. |
Added to introduction |
|
5B |
Include in the discussion section further hypotheses on the gap- perhaps the broadness of dementia of provider discomfort in making the referral. |
Further hypothesized on the gap in paragraph 2 of the discussion. |
|
5C |
Note whether this includes people who were provided information but declined the referral. |
The data do not include people who declined referrals; however, we do not know that caregivers accessed the resources and support services they were informed about. This has been added via a note under Tables 6, 7, 8 and 9. |
Reviewer 6 Report
Comments and Suggestions for Authors
First of all, I would like to congratulate you on your study as it contributes to the improvement of the care of teaching staff. However, it is necessary to include in the discussion section the future lines of research and practical implications found in the present study. In the summary, it is necessary to include the objective of the study and the most relevant conclusions.
Author Response
|
Reviewer 6 - Thank you for your helpful comments. We appreciate your time, expertise and suggestions. Below is a list of your comments and our responses/changes.
|
||
|
6A |
Include in the discussion section the future lines of research and practical implications found in the present study. |
Addressed in 2P and 2Q |
|
6B |
Include the objective of the study and the most relevant conclusions in the summary. |
Addressed in 1A
|
|
2P |
Elaborate more thoroughly on the specific implications of the results for clinical practice, health policies and future research. |
Expanded the third paragraph of the conclusion to include specific implications of the results for clinical practice, health policies and future research. |
|
2Q |
Provide more specific recommendations for future interventions and research |
Expanded the third paragraph of the conclusion to include specific recommendations for future interventions and research
|
|
1A |
Adjust abstract to include any major findings and conclusions. |
Added sentence to reflect major findings and conclusions to abstract. See 2 D below for more on abstract adjustment to include major findings. |
|
2D |
Include quantitative data within the abstract. 250 words |
See new results section recommended in 2E. The results include quantitative data from major findings in Results section as suggested by this reviewer and reviewer 1. |